# Models of Distal Arthrogryposis and Lethal Congenital Contracture Syndrome

**DOI:** 10.3390/genes12060943

**Published:** 2021-06-20

**Authors:** Julia Whittle, Aaron Johnson, Matthew B. Dobbs, Christina A. Gurnett

**Affiliations:** 1Department of Neurology, Washington University in St Louis, St Louis, MO 63130, USA; juliawhittle@wustl.edu; 2Department of Developmental Biology, Washington University in St Louis, St Louis, MO 63130, USA; anjohnson@wustl.edu; 3Paley Orthopaedic and Spine Institute, West Palm Beach, FL 33407, USA; mdobbs@paleyinstitute.org

**Keywords:** contracture, arthrogryposis, congenital

## Abstract

Distal arthrogryposis and lethal congenital contracture syndromes describe a broad group of disorders that share congenital limb contractures in common. While skeletal muscle sarcomeric genes comprise many of the first genes identified for Distal Arthrogyposis, other mechanisms of disease have been demonstrated, including key effects on peripheral nerve function. While Distal Arthrogryposis and Lethal Congenital Contracture Syndromes display superficial similarities in phenotype, the underlying mechanisms for these conditions are diverse but overlapping. In this review, we discuss the important insights gained into these human genetic diseases resulting from in vitro molecular studies and in vivo models in fruit fly, zebrafish, and mice.

## 1. Introduction

Arthrogryposis (arth = joint; grp = curved; osis = pathological state) describes a broad range of phenotypes consisting of multiple congenital joint contractures presenting at birth [1]. About 1 in 3000 live births presents with some form of arthrogryposis, many of which are nonprogressive and improve with physiotherapy. The core root of arthrogryposis is fetal akinesia, or lack of fetal movement, that results in contractures forming in the joints [1,2,3]. Movement is required for normal joint development; it influences the structure of the joints, as well as promoting cellular signaling that guides normal tissue development. Mechanical forces also influence bone morphology, affecting organization of chondrocytes, bone elongation, and differential growth, all affecting the shape of bones as they develop. Fetal akinesia impairs joint formation, which may lead to joint fusions. Furthermore, tension is required for normal tendon development, forming a connection between bone and muscle [4]. Arrested movement during development has significant impact on the formation of the skeleton, joints, muscle, and connective tissues.

The full range of joint movement in utero can be perturbed both intrinsically and extrinsically. Intrinsically, mutations affecting the muscle, bone, connective tissue, and neural system can affect the range of movement of joints. Currently, there are over 400 genes associated with arthrogryposis broadly, encapsulating a wide diversity of genes affecting different pathways, including genes associated with axon structure, circulatory development, or synaptic transmission [5]. Extrinsically, maternal disease or exposures, uterine space limitation, and decreased blood supply are also root causes for contraction defects [1,2]. Because joint motion is affected by many different systems, a wide range of issues during development can arrest joint motion.

A subset of arthrogryposis is described as Distal Arthrogryposis (DA), a group of genetically induced contractures that predominantly affect the joints of the distal limbs, including the hands, wrists, ankles, and feet. Clinically, the lower extremity manifestations commonly include clubfoot and vertical talus. There are currently 10 classifications of DA, including Sheldon-Hall syndrome (DA2B) and Freeman-Sheldon syndrome (DA2A) [6,7,8,9]. Freeman-Sheldon syndrome is considered the most severe form of DA, and also presents with facial contractures [9].

Currently, DA patients are offered supportive care to improve quality of life, including occupational therapy, physical therapy, and surgery [10]. While these treatments improve outcome for patients, they often fall short of complete restoration of range of motion in the joints and functionality. This strategy also fails to address underlying causes for DA, such as muscle weakness and impaired neurotransmission. Therefore, further investigation is necessary to understand the impact of disease variants which will allow us to determine the most effective treatment options for patients.

Lethal Congenital Contracture Syndromes (LCCS) are included in this review, as some of the same genes and disease mechanisms apply to this serious condition, which is typically fetal or neonatal lethal. LCCS presents with severe generalized contractures, along with many other typical features including incomplete lung development, and polyhydramnios [11]. In contrast to DA, which is most often inherited as an autosomal dominant condition, LCCS has only been described in the autosomal recessive state. Eleven subtypes have been described to date [10].

Various disease models have been developed to examine the mechanisms behind DA and to test therapeutic interventions (Table 1). Molecular and single-cell studies are useful for precisely examining the effects of DA-causing variants on the affected proteins and tissues. Access to human tissues, particularly from muscle biopsies, has facilitated molecular analysis for research, yet is clinically useful only in select cases [12,13]. In addition, protein modeling can help predict the impact of various amino acid substitutions on molecular interactions [14,15]. On the other hand, animal models are necessary to analyze the effect of single gene variants on organisms on scales larger than single cells. The effect of zygosity and gene dosage may also be better studied in animal models to assess interactions between normal and abnormal gene products. Animal models are also useful for studying experimental interventions that may improve patient quality of life and outcome, acting as stand-ins for potential human patients.

Models of human disease are rapidly becoming more sophisticated, with the ability to knock-in single nucleotide variants and create conditional (tissue specific or time-dependent) knockouts [16,17]. Loss-of-function alleles, which are often easier to generate, provide critical information about gene function, but may not fully explain autosomal dominant phenotypes in which gain-of-function or dominant negative effects can cause markedly different phenotypes. Conditional knockouts, while very helpful in defining gene function, rarely replicate the human phenotype in its entirety, but may be required when early lethality limits further study. These methods allow researchers to design models that more accurately represent these human conditions, and replicate pathogenic effects broadly or in specific tissues.

This review will examine genetic models of DA and LCCS, and the impact they have had in understanding the underlying pathophysiology. While we will describe both in vitro and in vivo approaches, we will focus primarily on vertebrate models, as these have the potential to provide insight into the multifaceted effects of disease variants on the multiple tissue types that contribute to these complex human phenotypes. We will also examine the current trajectory of DA research, and how these research strategies can help those afflicted by DA.

## 2. Muscle-Related Distal Arthrogryposis

### 2.1. MYH3

Missense mutations in *MYH3*, the earliest expressed embryonic myosin heavy chain gene that is predominantly expressed in myotubes destined to become fast-twitch myofibers [9], are strongly associated with DA clinically, and contribute to multiple subtypes including DA1, DA2A, and DA2B with varying degrees of severity [7,8,9,59,60]. *MYH3*-associated DA is almost always caused by single missense variants, as frameshift knockout or premature stop mutations are frequently observed in healthy population controls [9]. However, nonsense *MYH3* variants may contribute to autosomal recessive spondylocarpotarsal syndrome in the compound heterozygous state when presenting along with a missense allele [21], and have also been described with autosomal dominant spondylocarpotarsal syndrome [61]. DA-associated pathogenic variants cluster in the motor domain, but have also been found in the tail region of the protein [9]. Many of these missense variants are de novo, but some segregate in families with complete, or nearly complete penetrance [9]. *MYH3* appears to be one of the most common genes associated with DA, therefore various in vitro and in vivo studies, including protein modeling, cell models, and vertebrate studies, have been performed to elucidate the effects of *MYH3* variants on muscle function and the subsequent effects on the joints and skeleton.

#### 2.1.1. Biochemical and Cell Models for *MYH3*-Associated Distal Arthrogryposis

Single molecule and single cell studies are useful to examine the precise impact of a variant on protein function. The effects of amino acid substitutions are difficult to predict without mechanistic examination. Single-molecule studies facilitate understanding the effect of a missense variant on protein function and can later be translated into an understanding of how small mechanical differences affect tissues and whole body systems. Missense variants can be studied in human skeletal muscle biopsies. However, these are not routinely performed for DA diagnosis, which makes these studies challenging. To study this mechanistic link between DA phenotype and gene variant, Racca et al. performed contractility studies on isolated muscle cells and myofibrils derived from biopsied muscle tissue from DA2A patients [12]. They found that a DA2A-associated *MYH3* variant inhibited cross-bridge detachment, thereby slowing muscle relaxation and lowering force production. A later study replicated these results while examining multiple *MYH3* variants associated with DA2 [54]. In addition to slower actin-myosin detachment, ATP binding and ATPase activity were lower in variant MYH3 molecules.

Development of single cell models, in which *MYH3* variants are exogenously expressed or overexpressed from a plasmid or virus, have been limited by the difficulties of expressing large genes, like *MYH3*, in vitro. Other key challenges of in vitro modeling include the paucity of skeletal muscle cell lines other than C2C12 cells, and the propensity of muscle cells to form a syncytium. In addition, difficulties in differentiating human induced pluripotent stem cells (iPSCs) into skeletal muscle have also limited their use for arthrogryposis disease modeling. Likewise, because many features of DA are due to complex relationships between different cell types, co-cultures of muscle cells with tenocytes and bone may be required to recapitulate the human condition. Thus, many investigators have preferred to study DA genes in whole organisms.

#### 2.1.2. Invertebrate Models for *MYH3*-Associated Distal Arthrogryposis

*Drosophila melanogaster* (fruit flies) are useful tools for studying muscle function and myofibril assembly, particularly as introduction of single variants are traditionally simpler in this system compared to other models. *MYH3* and the *Drosophila* myosin heavy chain gene, *Mhc*, are highly conserved. *Drosophila* have the advantage of having only a single myosin heavy chain, which eliminates the possible obscuration of effects due to compensation by other myosin heavy chain analogs. Therefore, the effect of variants on protein function can be examined in a setting without other myosin heavy chain isoforms.

*Drosophila* transgenic models have been generated by overexpressing *Mhc* constructs containing DA variants [14,25]. Guo et al. predicted that a DA1 variant would perturb a hydrophobic interaction, while a DA2B mutation would introduce a hydrogen bond that was not present in the wild type. The effect of these predicted interactions was tested mechanistically in *Drosophila* models. Muscle fibers containing the DA alleles were extracted and found to have lower actin affinity, reduced power output, and increased stiffness, which may explain the motor deficits [14].

Morphological studies of *Drosophila* skeletal muscle expressing three DA2A *Mhc* transgenes (R672H, R672C, and T178I) showed branching and splitting defects, which were most severe in the R672C variant, which caused Z-discs to be split and malformed. In addition, Z-disc distance was shorter in the transgenic flies indicating an overall shortening of the sarcomeres, perhaps due to an enhanced contractile state of the myofibers. Presumably, the shortened sarcomeres observed in *Drosophila* contribute to the formation of contractures in human patients. Indeed, ATPase activity was reduced in these transgenic flies, leading to functional defects in muscle activity [25].

In studying the intact adult *Drosophila*, Guo et al. showed that the *Mhc*^F437I^ mutants had a much longer lifespan than *Mhc*^A234T^ mutants, consistent with the less severe phenotype of DA1 patients compared to DA2B patients [14]. In addition, the researchers found that both mutants displayed aberrant myofibril assembly, as well as misaligned sarcomere structure including distorted M and Z lines [14]. Again, this phenotype was more severe in A234T mutants than in the F437I mutants. *Mhc*^F437I^ mutants displayed essentially normal myofibers and sarcomeres, while *Mhc*^A234T^ mutants had small myofibers with disrupted morphology, as well as abnormal sarcomeres [14]. Das et al. also found that decreased climbing capability of adult flies also correlated with the phenotypic severity in humans [25].

Like *Drosophila*, *Caenorhabditis elegans* (*C. elegans*) has also been used to study myosin heavy chain genes [62]. There are many advantages of *C. elegans* and *Drosophila* for disease modeling, including large numbers of progeny, knowledge of ontogeny of individual cells, and ease of functional studies for drug screening. However, as described in Gil-Galvez et al., the evolutionary distance, and differences in number of myosin heavy chain genes between invertebrates and humans makes it difficult to determine whether the gene being studied has the same function, particularly in terms of spatial and temporal expression, as its human counterpart. Furthermore, Gil-Galvarez et al. caution against overexpression studies in *C. elegans* broadly, citing interference with muscle cell function overall [62]. The major drawback of invertebrate models is, quite obviously, the lack of skeletal structures which limits their use in understanding the complex relationship between muscle, nerve, and bone.

#### 2.1.3. Vertebrate Models for *MYH3*-Associated Distal Arthrogryposis

Germline loss of *Myh3* in mice results in altered muscle fiber size, fiber number, fiber type, and misregulation of genes, and adult *Myh3* null mice develop scoliosis [32]. However, the molecular defect may make these mice a better model for recessive *MYH3* spondylocarpotarsal synostosis syndromes than for autosomal dominant DA [21,61]. Notably, many patients with spondylocarpotarsal synostosis syndrome also have congenital contractures, which highlights the phenotypic overlap between DA and spondylocarpotarsal synostosis syndrome. Interestingly, *MYH3* was also shown to be expressed in bone, which the authors state may explain the effects of *MYH3* variants on both skeletal muscle and bone, particularly for patients with spondylocarpotarsal synostosis syndrome and bony fusions [61].

To more accurately model DA2A, Whittle et al. recently introduced one of the most common Freeman-Sheldon syndrome *MYH3* variants, R672H, into an analogous gene in zebrafish (*Danio rerio*) (*smyhc1*^R673H^) [16]. Zebrafish breed profusely and are cost-effective compared to mice. They also mature quickly, develop in vitro, and are transparent in the first few days of life, which facilitates imaging. Zebrafish are also vertebrates, making them more closely related to humans than *Drosophila* or *C. elegans*. Because two zebrafish lines were created, including *smyhc1* null and *smyhc1*^R673H^ lines, gene dosage effects were studied by examining the variant in the context of different zygosities. Indeed, *smyhc1*^R673H^ homozygotes displayed severe, early lethal phenotype compared to *smyhc1*^R673H^ heterozygotes, indicating that the *smyhc1*^R673H^ mutation acts as a hypermorph [16]. This result suggests human fetal lethality if a DA missense variant occurs in the homozygous state, which has not yet been described.

Zebrafish larvae harboring the *smyhc1*^R673H^ variant demonstrated severe notochord kinks [1], and adults had vertebral fusions that were similar to those seen in patients autosomal dominant spondylocarpotarsal synostosis due to *MYH3* variants. On histological examination, skeletal muscle showed severely shortened and misshapen muscle fibers. Similar to studies in *Drosophila*, the somite length was reduced in *smyhc1*^R673H^ mutants, consistent with shortening of the sarcomere.

A major advantage of zebrafish is the ease with which drugs can be administered for therapeutic investigations and drug screening. Based on knowledge that myosin ATPase inhibitors are now being evaluated to treat human cardiomyopathy due to similar variants in cardiac myosin genes [63], Whittle et al. preemptively treated embryos with para-aminoblebbistatin to prevent contractures from forming in larvae. Para-aminoblebbistatin inhibits myosin heavy chain ATPase activity, which chemically relaxed the skeletal muscle and prevented the curved phenotype of the treated *smyhc1* mutant fish (Figure 1) [16]. Previous molecular and single fiber studies predicted this mechanistic effect. Based on this experimental work, myosin ATPase inhibitors may be a viable avenue for *MYH3*-associated DA treatment, but will most likely require development of skeletal-muscle-specific inhibitors and treatment at an appropriately early developmental window.

### 2.2. MYBPC1 and MYBPC2

Strong evidence now exists linking variants in the slow skeletal muscle myosin binding protein C1 (*MYBPC1*) to dominantly inherited DA1 [30], DA2 [58], arthrogryposis multiplex congenita [28], myopathy with tremor [49], and, in the recessive state, to lethal congenital contracture syndrome LCCS4 [11].

Morpholino knockdown of *mybpc1* in zebrafish resulted in embryos with severe body curvature, as well as impaired motor excitation with defective myofibril organization and reduced sarcomere numbers [31]. Furthermore, overexpression of human *MYBPC1* DA1-associated variants in zebrafish resulted in hypermorphic effects with body curvature, decreased motor activity, and impaired survival. No effect was seen with overexpression of wild-type transcripts, suggesting that overexpression studies in zebrafish could be an efficient model for future functional testing of the human variants of uncertain clinical significance.

In contrast to *MYBPC1,* which is strongly implicated in human disease, the role of fast skeletal muscle myosin binding protein C2 (*MYBPC2*) in DA is less clear, as there is only a single report of *MYBPC*2 variants in DA patients in whom other known arthrogryposis gene variants were also observed, suggesting a possible role as a modifier [18]. Knockdown of *MYBPC2* with morpholino oligonucleotides produced a myopathic phenotype [37], but single variants have not yet been studied.

### 2.3. TPM2

*TPM2* variants cause a spectrum of phenotypes, including DA1, DA2, as well as nemaline myopathy and cap myopathy (reviewed in (Tajsharghi et al., 2012)) [64]. All are autosomal dominant with the exception of a pathogenic null variant identified in a consanguineous family with Escobar variant of multiple pterygium syndrome that was observed in the recessive state [65]. Biochemical studies were undertaken to study *TPM2* gain-of-function phenotypes, including in vitro motility assays, which showed variable effects on calcium sensitivity and tropomyosin flexibility [20,39]. In addition, *TPM2* was recently shown to have noncanonical roles other than its sarcomeric function, where it binds thin filament actin to regulate muscle contraction. In this work, *TPM2* directly regulated muscle morphogenesis by directing myotubes toward tendon attachment sites [56]. Muscle morphology was disrupted in both flies and zebrafish expressing DA1-associated *TPM2* variants, likely by causing myofiber hypercontraction (Figure 2).

### 2.4. TNNI2

While the function of the fast skeletal muscle Troponin I (*TNNI2*) has been described in flightless *Drosophila* models [53], only a single DA disease-associated missense variant has been modeled in mice, which accurately recapitulated the human disease [17]. However, the small body size of mice carrying the *TNNI2* DA variant could not be explained by the direct effect of the variant on skeletal muscle morphology or function. Rather, *TNNI2* was shown to be expressed in osteoblasts and chondrocytes of long bone growth plates, through which its effects on growth was predicted to occur. Therefore, like the studies described above for *MYH3*, this model provides evidence that some DA phenotypes may be directly attributable to expression in non-muscle tissue, such as bone.

### 2.5. TNNT3

DA-associated variants in fast skeletal muscle Troponin T (*TNNT3*) are also dominantly inherited missense variants, and therefore, like those in many genes described previously, cause disease through a gain-of-function manner and therefore cannot be adequately modeled using a simple knockout approach. Therefore, while knockout approaches have shown a critical function of *TNNT3* during vertebral development [34], no models with disease-specific missense variants have been generated, and future studies are needed.

### 2.6. MYLPF

Exome sequencing recently identified *MYLPF*, a phosphorylatable fast skeletal muscle regulatory light chain, as a cause of DA [23]. Some affected individuals were homozygous for rare variants in the gene, while other individuals have autosomal dominant disease, a finding similar to what was described for *MYH3*-related disorders. However, unlike *MYH3*-related disorders, the phenotypes for *MYLPF* autosomal dominant and recessive conditions are apparently indistinguishable. Protein modeling of *MYLPF* alleles suggested that the autosomal dominant pathogenic variants cause disease through their direct interaction with myosin, while the recessive alleles only indirectly affect the interaction with myosin [23].

Previously described *mylpf* knockout mice did not develop either fast or slow type skeletal muscle mass, which resulted in early death before or after delivery, presumably due to respiratory failure [66]. Similarly, an individual with recessive *MYLPF*-associated DA1 was found to have absent skeletal muscle in an amputated foot, which could best be described as amyoplasia. The recessive *MYLPF* pathogenic variant in this individual was hypothesized to be hypomorphic. Therefore, to further model hypomorphic alleles of *MYLPF*, a zebrafish *mylpfa* mutant was characterized. Because zebrafish have 2 *MYLPF* genes, *mylpfa* and *mylpfb*, the more prominently expressed gene (*mlfpfa*) was chosen to model hypomorphic *MYLPF* alleles [23]. Of note, zebrafish had an evolutionary genome duplication event that has resulted in many human genes being represented twice in the zebrafish genome. Some of these genes have subsequently evolved to occupy additional temporal or spatial expression patterns and/or adopted new developmental roles. While this duplication event may complicate identification of the relevant human gene, gene duplications can also be an advantage, allowing genes to be studied whose knockout is early embryonic lethal in other species.

*Mylpfa* zebrafish null mutants were found to have paralyzed pectoral fins, an impaired escape response, and consistently lower trunk muscle force compared to wildtype [23]. In in vitro studies, myosin extracted from *mylpfa* mutant larvae propelled significantly more slowly than their wild type protein. Skeletal muscle fibers were also found to degenerate in mutant larvae, collapsing and losing structure, developing membrane abnormalities, indicating that *mylpf* is necessary to maintain cellular integrity in muscle cells [23]. Thus, *MYLPF* may be unique among the DA genes in also causing amyoplasia, which may have important implications for personalized therapeutic strategies.

## 3. Neural-Related Distal Arthrogryposis

### 3.1. PIEZO2

DA is not only associated with muscle proteins. Whole genome sequencing was performed on individuals with DA5, in which arthrogryposis occurs in combination with ptosis, ophthalmoplegia, and facial dysmorphism, and gain of function variants (I802F and E2727del) were discovered in *PIEZO2* [24], a mechanosensitive cation channel responsible for mediating cation currents in primary sensory neurons [24,40,55]. Because of their mechanosensitivity, these channels are also termed “stretch-activated ion channels”. *PIEZO2* was found to affect the skeleton non-autonomously in mice. Loss of *PIEZO2* specifically in skeletal tissue did not affect bone development; however, loss of gene function in proprioceptive neurons caused spine malalignment [67]. This suggests both that the neural system is necessary in maintaining normal skeletal development, and that *PIEZO2* is a critical gene in this process.

To test the mechanistic effects of these pathogenic variants, one group recently transfected human embryonic kidney cells with wild type *PIEZO2*, or *PIEZO2* with missense variants encoding I802F, or E2727del [24]. Both of these disease-associated variants caused the channel to recover more quickly from inactivation and resulted in increased channel activity following a mechanical stimulus. This supports the hypothesis that DA5 is caused by gain of function mutations that alter mechanosensory nerves. Although additional studies are needed, overstimulation may directly affect the neuromuscular pathway that controls muscle tone in developing fetuses, perhaps causing near-constitutive contractions that constrain the developing joint.

### 3.2. ECEL1

Variants in *ECEL1*, which encodes the endothelin-converting enzyme like-1, were identified in the recessive state in several families with DA5D, a rare form of arthrogryposis in which affected individuals have contractures as well as distinctive facial features and ptosis [41]. *ECEL1*, which is expressed in brain and nerve, is required for post-natal development in mice. Loss of *Ecel1* in mice results in abnormal terminal branching of motor neurons at the skeletal muscle endplate [68]. The mechanism by which *ECEL1* directs motor neuron branching is currently unknown; however, the resulting contractures in patients with *ECEL1* variants were proposed to be caused by a similar mechanism to those caused by genes such as *CHRNG* that causes multiple pterygium syndrome that impairs neurotransmission at the neuromuscular junction [69].

## 4. Lethal Congenital Contracture Syndrome

In contrast with DA, which are more common and often autosomal dominant, Lethal Congenital Contracture Syndromes (LCCSs) are a group of rare autosomal recessive forms of arthrogryposis. LCCS are characterized by lack of fetal movement (akinesia), micrognathia, incomplete lung development, polyhydramnios, characteristic contractures of the limbs (clubfoot, hyperextended knees, elbow and wrist flexion contractures) and motoneuron degeneration. Eleven subtypes of LCCS have been characterized. However, there are likely to be many more genes that result in these conditions as more genetic studies are performed on products of conception due to spontaneous abortion or stillbirth. LCCS is more common in communities with high rates of consanguinity consistent with the recessive inheritance pattern. Variable expression of LCCS phenotypes may be due to residual gene function in patients with missense variants or modifier genes. The recessive phenotypes of LCCS have made them more amenable to study by complete knockdown of gene expression.

### 4.1. Nuclear mRNA Export (GLE1, ERBB3, and PIP5K1C)

The first three LCCS subtypes may all act through a similar pathway by supporting nuclear mRNA export. LCCS type 1 (LCCS1) is caused by mutations in *GLE1* RNA Export Mediator (*GLE1*), a regulator of post-transcriptional gene expression [70]. *GLE1* acts as an mRNA export factor, as well as by mediating translation initiation and termination [15]. In mice, in situ hybridization showed marked expression in the neural tube of 11 dpf embryos, specifically in the ventral portion from which motoneurons generate [70]. In zebrafish, the gene is expressed prominently in the central nervous system during development [33].

A mutation in *GLE1*, Fin_Major_, has been linked to LCCS1 by causing a splice-site mutation that results in a 3 amino acid insertion in the coiled-coil domain [70]. The coiled-coil domain is required for the protein to self-associates to form oligomers, and one group examined the effect of the Fin_Major_ mutation on polypeptide self-association in vitro and in vivo [29]. Both in vitro and in living cells, the GLE1 protein self-aggregated, and Fin_Major_ mutant oligomers were malformed. In human cell culture and in the yeast model, these malformed oligomers were found to perturb mRNA export from the nucleus [29].

Because the Fin_Major_ mutation reduced function of the GLE1 protein in mRNA transport, *gle1* knockdown and knockouts were studied in zebrafish to understand its effects on development. Knockouts developed with small eyes and underdeveloped jaws and pectoral fins [33]. Cell death was also observed in the head and spinal cord, and there were fewer motoneurons than in wild type fish. Motoneurons also exhibit aberrant branching that worsened with age. Maternal *gle1* mRNA is loaded into the yolk sac of oocytes, where it contributes to zebrafish embryogenesis; therefore, morpholino oligonucleotides were also used to knock down expression of the mRNA in embryos. This exacerbated the phenotype, with CNS cell death becoming apparent earlier in development, at 1 dpf, which suggests an important role for *gle1* for early development. Notably, this phenotype is rescued in morphants injected with human wild type *GLE1*, but not when injected with the Fin_Major_ allele [33]. Thus, this zebrafish model may be a viable tool for screening and determining the pathogenicity of human alleles.

LCCS2 is due to loss-of-function mutations in Erb-B2 Receptor Tyrosine Kinase 3 (*ERBB3*), which encodes HER3, a known modulator of the phosphatidylinositol pathway [44]. Interestingly, variants in LCCS3 were found to be due to variants in Phosphatidylinositol-4-Phosphate 5-Kinase Type 1 (*PIP5K1C*), which encodes the enzyme PIPK-gamma of the phosphatidylinositol pathway [43]. Nouslainen et al. realized that both *ERBB3* and *PIP5K1C* are involved in the synthesis of inositol hexakisphosphate, which binds directly to yeast *Gle1*, activating *Dbp5* for mRNA transport [70]. Because Gle1 is expressed in the neural tube during development, pathogenic variants in this gene can be devastating to development of the nervous system, as *Gle1* is integral to mRNA transport [70].

### 4.2. Peripheral Nerve (CNTNAP1, ADGRG6, GLDN)

The genes responsible for LCCS7, LCCS9, and LCCS11 are all highly expressed in peripheral nerves and required for proper peripheral nerve function. *Contactin Associated Protein 1* (*CNTNAP1)*, which causes LCCS7, is a contactin-associated protein that is required for localization of the paranodal junction proteins contactin and neurofascin. *CNTNAP1* is also required for the normal spatial expression patterns of neuronal sodium and potassium channels [19]. Likewise, the causative gene for LCCS11, *gliomedin (GLDN)*, is a ligand for neurofascin and Nrcam, which are axonal immunoglobulin cell adhesion molecules critical for association with sodium channels at the nodes of Ranvier [71]. *Adhesion G Protein Coupled Receptor G6 (ADGRG6)*, which is also known as GPR126, is required for normal Schwann cell development. Thus, defects in all three genes likely result in similar peripheral nerve dysfunction at very early stages in development that leads to the LCCS phenotype.

## 5. Conclusions

Many techniques and organisms have been used for modeling arthrogryposis, each of which provides complementary information that is essential for understanding basic mechanisms and will yield translational benefits to human patients. There is an expanding list of genes that are associated with limb contractures, as one of many clinical features, beyond those discussed in this review article. Other genes are yet to be discovered, and disease models are often needed to provide evidence of causality. Furthermore, as exome sequencing becomes standard care, disease models may be helpful to facilitate variant interpretation. However, it will be essential to develop more efficient methods for introducing and studying large numbers of individual variants.

Although most genes responsible for distal arthrogryposis and LCCS are skeletal muscle sarcomeric genes or genes critical for neuronal function and neuromuscular transmission, crucial aspects remain to be established using disease models. It is important to determine whether common pathways and mechanisms supported by the genetic data will predict a unifying approach to therapy. Furthermore, now that gene therapies are becoming viable treatment mechanisms, where and when the defect needs to be corrected to prevent development of the DA or LCCS phenotype needs to be elucidated. Disease models will be essential to improve treatment for these challenging disorders.

## Figures and Tables

**Figure 1 genes-12-00943-f001:**
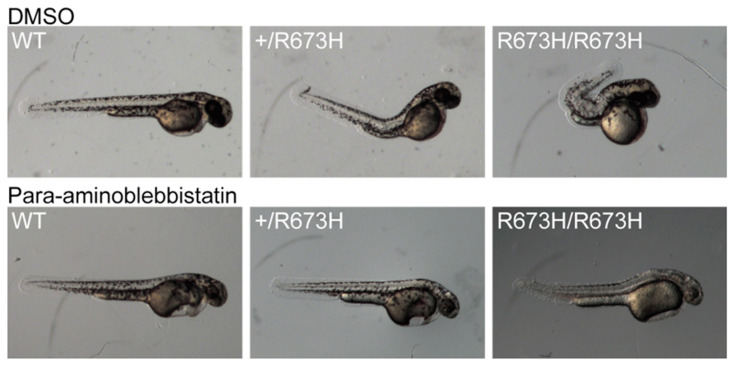
The curved spinal phenotype associated with both *smyhc1*^+/R673H^ and *smyhc1*^R673H/R673H^ genotypes is normalized with the myosin inhibitor para-aminoblebbistatin. Embryos were treated from 24–48 hpf and photographed at 48 hpf. Treated embryos are shown below DMSO treated controls. Unlike the newer myosin inhibitors that are being developed, para-aminoblebbistatin has many toxic effects, including lethal cardiac edema, which limits its use as a human therapeutic. These images are similar to those published in [16].

**Figure 2 genes-12-00943-f002:**
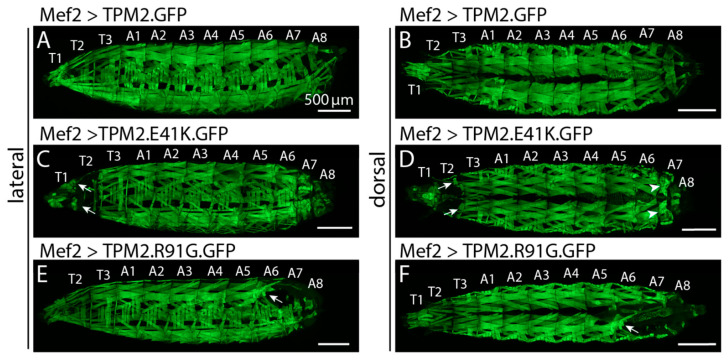
DA associated *TPM2* variants cause muscle phenotypes in *Drosophila*. Confocal micrographs of live L3 larva that express GFP-tagged TPM2 variants in skeletal muscles (body wall muscles). Mef2.Gal4 was used to activate UAS.TPM2 transgenes. Lateral and dorsal views are shown for each genotype. (**A**,**B**) Larva that express TPM2.GFP showed normal muscle histology. Larva that expresses TPM2.E41K.GFP (**C**,**D**) or TPM2.R91G.GFP (**E**,**F**). GFP have rounded myofibers that appear to result from internal tears (arrows; note affected muscles remain associated with tendons at segment boundaries) and shortened segments that could be due to hypercontractile muscles (arrowheads). Thoracic segments (T1–T3) and abdominal segments (A1–A8) are labeled. Scale bars, 500 mM. Previously unpublished data.

**Table 1 genes-12-00943-t001:** List of genes and associated conditions and models of distal arthrogryposis (DA) and lethal congenital contracture syndrome (LCCS) used for study. Autosomal dominant (AD), Autosomal recessive (AR) [9,11,12,14,16,17,18,19,20,21,22,23,24,25,26,27,28,29,30,31,32,33,34,35,36,37,38,39,40,41,42,43,44,45,46,47,48,49,50,51,52,53,54,55,56,57].

Gene	Full Name	Disorder	Inheritance Pattern	Modeled in	Source Human	Models of Disease Source
ARTHROGRYPOSIS						
MYH3	Myosin, Heavy Polypeptide 3, Skeletal Muscle, Embryonic	DA1, DA2A, DA2B, DA8, Spondylocarpotarsal Syndrome	AD, AR	Zebrafish, Cell, Biochemical	Toydemir et al., 2006b [9]; Chong et al., 2015 [57]; Cameron-Christie et al., 2019 [21]	Racca et al., 2015 [12]; Walklate et al., 2016 [54]; Wang et al., 2019 [55]; Whittle et al., 2020 [16]; Guo et al., 2020 [14]; Das et al., 2019 [25]
TPM2	Tropomyosin 2	DA1, Cap Myopathy, Nemaline Myopathy	AD, AR	Drosophila, Biochemical	Sung et al., 2003 [50]	Williams et al., 2015 [56]; Borovikov et al., 2017 [20]; Matyushenko & Levitsky, 2020 [39];
MYLPF	Myosin Regulatory Light Chain 2, Skeletal Muscle Isoform	DA1, DA2B	AD, AR	Zebrafish	Chong et al., 2020 [23]	Chong et al., 2020 [23]
MYBPC1	Myosin-Binding Protein C, Slow-Type	DA1, DA2, LCCS4	AD	Zebrafish	Gurnett et al., 2010 [30]; Li et al., 2015 [58]; Ekhilevitch et al., 2016 [28]; Shashi et al., 2019 [49];	Ha et al., 2013 [31]
MYBPC2	Myosin-Binding Protein C, Fast-Type	DA (unspecified)	AD	Zebrafish	Bayram et al., 2016 [18]	Li et al., 2016 [37]
TNNT3	Troponin T3, Fast Skeletal Type	DA2B	AD, AR	Mouse	Sung et al., 2003 [50]; Sandaradura et al., 2018 [48]	Ju et al., 2013 [34]
TNNI2	Troponin I2, Fast Skeletal Type	DA2B	AD	Mouse, Drosophila	Sung et al., 2003 [50]	Zhu et al., 2014 [17]; Vigoreaux, 2001 [53]
PIEZO2	Piezo Type Mechanosensitive Ion Channel Component 2	DA3, DA5	AR	Cell	McMillin et al., 2014 [40]	Coste et al., 2013 [24]; McMillin et al., 2014 [40]
ECEL1	Endothelin Converting Enzyme Like 1	DA5 (or DA5D)	AR	-	McMillin et al., 2013 [41]	-
MYH8	Myosin, Heavy Polypeptide 8, Skeletal Muscle, Fetal	DA7	AD	-	Toydemir et al., 2006a; [51] Veugelers et al., 2004 [52]	-
LETHAL CONGENITAL CONTRATURE SYNDROME						
GLE1	GLE1 RNA Export Mediator	LCCS1	AR	Zebrafish, Cell, Biochemical	Jao et al., 2012 [33]	Folkmann et al., 2013 [29]; Jao et al., 2012 [33]
ERBB3	ERB-B2 Receptor Tyrosine Kinase 3	LCCS2	AR	Mouse	Narkis et al., 2007 [44]	Riethmacher et al., 1997 [47]
PIP5K1C	Phosphatidylinositol 4-Phosphate 5-Kinase, type 1, gamma	LCCS3	AR	Mouse	Narkis et al., 2007 [43]	DiPaolo et al., 2004 [26]
MYBPC1	Myosin-Binding Protein C, Slow-Type	LCCS4, DA1, DA2	AD, AR	Zebrafish	Markus et al., 2012 [11]	Ha et al., 2013 [31]
DNM2	Dynamin, 2	LCCS5, Centronuclear Myopathy, CMT2M, CMT Intermed	AD, AR	Mouse	Koutsopoulos et al., 2013 [35]	Durieux et al., 2010 [27]; Koutsopoulos et al., 2013 [35]
ZBTB42	Zinc finger-and BTB Domain-containing Protein 42	LCCS6	AR	Zebrafish	Patel et al., 2014 [45]	Patel et al., 2014 [45]
CNTNAP1	Contactin-associated protein 1	LCCS7, Congenital Hypomyelinating Neuropathy	AR	Mouse	Laquerriere et al., 2014 [36]	Bhat et al., 2001 [19]
ADCY6	Adenylyl cyclase 6	LCCS8	AR	Zebrafish	Laquerriere et al., 2014 [36]	Laquerriere et al., 2014 [36]
ADGRG6	Adhesion G-protein coupled receptor G6 or GPR126	LCCS9	AR	Zebrafish	Ravenscroft et al., 2015 [46]	Monk et al., 2009 [42]
NEK9	Nima-related kinase 1	LCCS10	AR	-	Casey et al., 2016 [22]	-
GLDN	Gliomedin	LCCS11	AR	-	Maluenda et al., 2016 [38]	-

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
