# Peer review of "Models of Distal Arthrogryposis and Lethal Congenital Contracture Syndrome"

_genes, 2021, doi:10.3390/genes12060943_

Round 1

Reviewer 1 Report

Very well written article on a complex topic. This review article nicely pulled together many pieces of information, helping to flesh out the puzzle of the etiology of distal arthrogryposis, as well as the lethal congenital contracture syndromes. The choice of pairing these two conditions in this one paper were not quite explained, as they only have one gene in common (MYBPC1). I suspect that the authors realized that their review topic was extremely wide and wanted to find two topics on which to focus, despite their lack of similarities. They do mention in their introduction that LCCS "are included in this review as some of the same genes and disease mechanisms apply". As such, I would suggest that they expand their conclusion to more thoroughly develop that thought (i.e. how the pathways of development of DA and LCCS, including the similarities of their fetal akinesia, are similar, and give greater insight to the pathogenesis of AMC in general).

There is little for me to comment on in regards to the content of this paper. I learned a lot by reading it. Two things that I hope the authors could touch on

1) we think of the DAs in general as being abnormalities of fast twitch muscle functioning or development, which explains the greater involvement of the hands and feet. Can the authors shed light on why MYH3, more of a ubiquitous muscle protein also leads to more distal effects, rather than to more generalized effects?

2) The authors discussed MYH3 mutations in regards to the DAs 1, 2A and 2B, but did not mention DA8, autosomal dominant pterygium syndrome. Could they touch on this in some way, ever to say that this was outside the scope of their paper for whatever reason?

Author Response

Very well written article on a complex topic. This review article nicely pulled together many pieces of information, helping to flesh out the puzzle of the etiology of distal arthrogryposis, as well as the lethal congenital contracture syndromes. The choice of pairing these two conditions in this one paper were not quite explained, as they only have one gene in common (MYBPC1). I suspect that the authors realized that their review topic was extremely wide and wanted to find two topics on which to focus, despite their lack of similarities. They do mention in their introduction that LCCS "are included in this review as some of the same genes and disease mechanisms apply". As such, I would suggest that they expand their conclusion to more thoroughly develop that thought (i.e. how the pathways of development of DA and LCCS, including the similarities of their fetal akinesia, are similar, and give greater insight to the pathogenesis of AMC in general).

There is little for me to comment on in regards to the content of this paper. I learned a lot by reading it. Two things that I hope the authors could touch on

1) we think of the DAs in general as being abnormalities of fast twitch muscle functioning or development, which explains the greater involvement of the hands and feet. Can the authors shed light on why MYH3, more of a ubiquitous muscle protein also leads to more distal effects, rather than to more generalized effects?

            Thank you for your comment- we have added a line at 99 to state that MYH3 is expressed in muscle cells that will become fast-twitch.

2) The authors discussed MYH3 mutations in regards to the DAs 1, 2A and 2B, but did not mention DA8, autosomal dominant pterygium syndrome. Could they touch on this in some way, ever to say that this was outside the scope of their paper for whatever reason?

            DA8 has been added to the table to reflect its causation by MYH3 mutations.

Reviewer 2 Report

This manuscript does a good job summarizing what recent genetically modified cell and animal models have taught us about arthrogryposes and lethal congenital contractures. It will make an important addition to the literature. However, I recommend that the authors distinguish different arthrogryposes and contractures much more clearly; it would also be improved by a major overhaul in how references are handled.

Major comments:

In the abstract, this review promises to show that arthrogryposes are caused by mutation in many genes not expressed in muscle. However, only two non-muscular genes are listed, both of which cause one specific type of distal arthrogryposis (DA5), and they both affect peripheral nerves at a point very close to muscle. Thus, there isn’t yet evidence that non-muscular genes can cause DAs generally – simply that muscle-adjacent processes can cause DA5 specifically. The authors also include a section about lethal congenital contracture syndrome. However, key features of LCCS are independent of limb contractures and both the genes and alleles underpinning this more-severe condition can differ from those causing DA. It makes sense for the authors to cover both DAs and LCCs in one review, but some more nuance would help speaking about these conditions together. This may require a little more space in both the abstract and discussion.

Likewise, this work claims to cover arthrogryposes broadly, but really is focused on genetic models of arthrogryposes, while skipping over work done on non-genetic causes. The authors could circumvent this last issue by stating clearly and explicitly that this review specifically covers genetic models of arthrogryposis.

Finally,  this manuscript has trouble with citations. For instance, the introduction contains a stretch of three paragraphs with no citations at all (Lines 52-80). In many other places it is unclear where the source of information comes from – where the material is drawn from that source cited several sentences earlier or later. In a few cases, the review discusses specific manuscripts and neglects to cite them entirely. Thus, as written, it would be difficult for readers to use this review to understand the literature.

Minor comments:

Line 57: It’s not clear what the authors mean by “the various systems of the body”. ‘Animal models provide critical insights about how mutations affect organisms on scales larger than cells’ ?

Lines 52-80 are completely lacking in references.

Lines 68-70 indicates that conditional knockouts rarely replicate human phenotypes. Such a statement is misleading, because conditional knockouts often do replicate aspects of human phenotypes and have been immensely useful for understanding human disease.

Line 85: Rephrase to clarify what is meant by “in a graded fashion”

References missing in lines 90-103.

Line 93 needs to be more specific than “various in vitro and in vivo studies have been performed”

Line 98 reads “effect amino acid substitutions” but should read “effect of…”

Line 129-130 states that ‘several groups have generated’ but only provide two citations.

Line 140-141 needs revision. Currently states: “Presumably, this shortening is related to the formation of contractures in DA patients.” Please revise this to a falsifiable statement about how the Drosophila phenotype is related to human contractures.

Line 147: What is “skewed sarcomere structure”? Revise for clarity. Likewise, clarify what is meant by ‘abnormal sarcomeres’

References are not clearly positioned in  lines 156-152 and a few other spots.

Line 152: I think the authors mean ‘phenotypic severity’, or ‘symptomatic severity’, rather than ‘mutational severity’, especially since they are referring to humans.

Overall, this review emphasizes the positive aspects of invertebrate animal models. So, it is strange to see C. elegans studies dismissed in lines 154-157. Gil-Gálvez et al., cautioned against using overexpression constructs to study disease variants because of ill-effects caused  by myosin overexpression – not a fundamental flaw in C. elegans itself. The same critique could potentially apply to mammalian myosin overexpression. Please rectify and incorporate references for the recent work using C. elegans to understand arthrogryposis gene variants. The authors also point out that C. elegans has a different number of Myosin heavy chain genes than humans; however, the same is true in other model organisms, so I don’t think this is a good road to travel down.

Typo in line 164 .‘may be a better’

Line 172: Whittle et al introduced MYH3 variants into an embryonic myosin heavy chain, but not the (only) analogous zebrafish gene. Your 2020 study was both clever and thought provoking - I only encourage you to soften this one word: the becomes an.

Resolution of figure 1 should be improved. Particularly, the text “DMSO” and “Para-aminoblebbsitatin” appears grainy.

Sentence in line 215-217 is unclear- could revise the word ‘this’ (216) to disambiguate.

Figure 2 legend states that this is unpublished data. However, it will be published when this article is published. Please either remove the statement (to match Figure 1) or state “previously unpublished”. Also, revise Figure 2 legend to explicitly mention panels E, F.

Line 265-273 alludes to a recent study about MYLPF causing DA, but the study itself is not cited until a paragraph later.

Piezo2 is traditionally thought to be a neural gene, but only recently (Assaraf 2020; PMID 31435011) has it been shown that loss of Piezo2 in neurons (not skeleton) can cause muscle/joint disease. This reference may aid your discussion of Piezo2 and integrate how this may influence understanding of DA5.

Line 309 appears to have a typo: “in increased in channel”. 

Line 335: Sometimes null alleles cause variably expressive phenotypes, so one cannot presume that the missense variation is due to residual function of the affected gene.

Line 337: It’s strange to state that LCCSs are more amenable to study by developmental biologists in a review with a hefty section about study of DA by developmental biologists.

Lines 356-376 have zero references and there is no way to infer what the references should be using context.

The sentence in lines 400-402 implies that the future of DA study lies in some undisclosed approach to study the effects of hundreds of gene variants at once rather than in living genetic models. This seems out of sync with the rest of the paper.

Author Response

This manuscript does a good job summarizing what recent genetically modified cell and animal models have taught us about arthrogryposes and lethal congenital contractures. It will make an important addition to the literature. However, I recommend that the authors distinguish different arthrogryposes and contractures much more clearly; it would also be improved by a major overhaul in how references are handled.

Major comments:

In the abstract, this review promises to show that arthrogryposes are caused by mutation in many genes not expressed in muscle. However, only two non-muscular genes are listed, both of which cause one specific type of distal arthrogryposis (DA5), and they both affect peripheral nerves at a point very close to muscle. Thus, there isn’t yet evidence that non-muscular genes can cause DAs generally – simply that muscle-adjacent processes can cause DA5 specifically. The authors also include a section about lethal congenital contracture syndrome. However, key features of LCCS are independent of limb contractures and both the genes and alleles underpinning this more-severe condition can differ from those causing DA. It makes sense for the authors to cover both DAs and LCCs in one review, but some more nuance would help speaking about these conditions together. This may require a little more space in both the abstract and discussion.

            Thank you for the comment. The abstract and conclusions sections hav been edited to bridge this gap more intuitively.

Likewise, this work claims to cover arthrogryposes broadly, but really is focused on genetic models of arthrogryposes, while skipping over work done on non-genetic causes. The authors could circumvent this last issue by stating clearly and explicitly that this review specifically covers genetic models of arthrogryposis.

            The abstract has been edited to specify the use of genetic models.

Finally,  this manuscript has trouble with citations. For instance, the introduction contains a stretch of three paragraphs with no citations at all (Lines 52-80). In many other places it is unclear where the source of information comes from – where the material is drawn from that source cited several sentences earlier or later. In a few cases, the review discusses specific manuscripts and neglects to cite them entirely. Thus, as written, it would be difficult for readers to use this review to understand the literature.

            Citations have been added to more accurately reflect the literature.

Minor comments:

Line 57: It’s not clear what the authors mean by “the various systems of the body”. ‘Animal models provide critical insights about how mutations affect organisms on scales larger than cells’ ?

            This has been clarified.

Lines 52-80 are completely lacking in references.

            References have been added in this section.

Lines 68-70 indicates that conditional knockouts rarely replicate human phenotypes. Such a statement is misleading, because conditional knockouts often do replicate aspects of human phenotypes and have been immensely useful for understanding human disease.

            We have added the phrase “in its entirety” to clarify.

Line 85: Rephrase to clarify what is meant by “in a graded fashion”

            This has been rephrased.

References missing in lines 90-103.

            References have been added.

Line 93 needs to be more specific than “various in vitro and in vivo studies have been performed”

            Specifi examples have been added.

Line 98 reads “effect amino acid substitutions” but should read “effect of…”

            This has been amended.

Line 129-130 states that ‘several groups have generated’ but only provide two citations.

            The wording has been changed.

Line 140-141 needs revision. Currently states: “Presumably, this shortening is related to the formation of contractures in DA patients.” Please revise this to a falsifiable statement about how the Drosophila phenotype is related to human contractures.

            This statement has been revised.

Line 147: What is “skewed sarcomere structure”? Revise for clarity. Likewise, clarify what is meant by ‘abnormal sarcomeres’

            This has been clarified.

References are not clearly positioned in  lines 156-152 and a few other spots.

            References has been clarified.    

Line 152: I think the authors mean ‘phenotypic severity’, or ‘symptomatic severity’, rather than ‘mutational severity’, especially since they are referring to humans.

            This has been changed.

Overall, this review emphasizes the positive aspects of invertebrate animal models. So, it is strange to see C. elegans studies dismissed in lines 154-157. Gil-Gálvez et al., cautioned against using overexpression constructs to study disease variants because of ill-effects caused  by myosin overexpression – not a fundamental flaw in C. elegans itself. The same critique could potentially apply to mammalian myosin overexpression. Please rectify and incorporate references for the recent work using C. elegans to understand arthrogryposis gene variants. The authors also point out that C. elegans has a different number of Myosin heavy chain genes than humans; however, the same is true in other model organisms, so I don’t think this is a good road to travel down.

            This paragraph has been changed to more clearly reflect the drawbacks of invertebrate models on DA phenotypes/

Typo in line 164 .‘may be a better’

            This has been changed.

Line 172: Whittle et al introduced MYH3 variants into an embryonic myosin heavy chain, but not the (only) analogous zebrafish gene. Your 2020 study was both clever and thought provoking - I only encourage you to soften this one word: the becomes an.

            This has been changed.

Resolution of figure 1 should be improved. Particularly, the text “DMSO” and “Para-aminoblebbsitatin” appears grainy.

            The figure has been rendered at a higher resolution.

Sentence in line 215-217 is unclear- could revise the word ‘this’ (216) to disambiguate.

            This has been changed.

Figure 2 legend states that this is unpublished data. However, it will be published when this article is published. Please either remove the statement (to match Figure 1) or state “previously unpublished”. Also, revise Figure 2 legend to explicitly mention panels E, F.

            This has been revised.

Line 265-273 alludes to a recent study about MYLPF causing DA, but the study itself is not cited until a paragraph later.

            The citation has been moved.

Piezo2 is traditionally thought to be a neural gene, but only recently (Assaraf 2020; PMID 31435011) has it been shown that loss of Piezo2 in neurons (not skeleton) can cause muscle/joint disease. This reference may aid your discussion of Piezo2 and integrate how this may influence understanding of DA5.

            This reference has been added.

Line 309 appears to have a typo: “in increased in channel”. 

            This has been changed.

Line 335: Sometimes null alleles cause variably expressive phenotypes, so one cannot presume that the missense variation is due to residual function of the affected gene.

            This statement has been changed.

Line 337: It’s strange to state that LCCSs are more amenable to study by developmental biologists in a review with a hefty section about study of DA by developmental biologists.

            The wording has been clarified.

Lines 356-376 have zero references and there is no way to infer what the references should be using context.

            References have been added.

The sentence in lines 400-402 implies that the future of DA study lies in some undisclosed approach to study the effects of hundreds of gene variants at once rather than in living genetic models. This seems out of sync with the rest of the paper.

                This has been clarified.

Reviewer 3 Report

Dear Author:

This is a very useful approach but you fail to put it into the larger picture of arthrogryposis where there are over 400 genes as recorded by Kiefer and Hall in 2019. 

Nor do you emphasize how very complex movement at different stages of development actually is. And all the multiple pathways involved   page 1 line 24: "Movement is required for normal joint development; it forms..." change "forms" to "influences"   page 1 line 31 "...wide range of issues during development can arrest this motion." change "this" to "joint"

Author Response

This is a very useful approach but you fail to put it into the larger picture of arthrogryposis where there are over 400 genes as recorded by Kiefer and Hall in 2019. 

Thank you, this has been added for extra context in the introduction.

Nor do you emphasize how very complex movement at different stages of development actually is. And all the multiple pathways involved   page 1 line 24: "Movement is required for normal joint development; it forms..." change "forms" to "influences"   page 1 line 31 "...wide range of issues during development can arrest this motion." change "this" to "joint"

These wording changes have been made and the ways in which movement affects skeletal and joint development have been expanded upon in the introduction.

Round 2

Reviewer 2 Report

The cleaned up manuscript looks great.  This review will be an excellent resource for both experts and people new to the field.